# Regional, Age, and Sex Patterns of Hepatitis C Virus Infection in Russia: Insights from a 42,000-Participant Serosurvey

**DOI:** 10.3390/v17121529

**Published:** 2025-11-21

**Authors:** Victor A. Manuylov, Vladimir A. Gushchin, Vladimir P. Chulanov, Olga V. Isaeva, Denis A. Kleymenov, Andrei A. Pochtovyi, Elena P. Mazunina, Evgeniia N. Bykonia, Irina N. Tragira, Yana V. Simakova, Sergey V. Netesov, Artem P. Tkachuk, Tatyana A. Semenenko, Alexander L. Gintsburg, Karen K. Kyuregyan, Mikhail I. Mikhailov

**Affiliations:** 1Gamaleya National Research Center for Epidemiology and Microbiology, 123098 Moscow, Russia; wowaniada@yandex.ru (V.A.G.); mne10000let@yandex.ru (D.A.K.); a.pochtovyi@yandex.ru (A.A.P.); mazuninaelenka@yandex.ru (E.P.M.); evgeniya_bikonya@mail.ru (E.N.B.); tragira@gamaleya.org (I.N.T.); y.v.simakova@gmail.com (Y.V.S.); artem.p.tkachuk@gmail.com (A.P.T.); semenenko@gamaleya.org (T.A.S.); gintsburg@gamaleya.org (A.L.G.); 2Laboratory of Bionanotechnology, Microbiology and Virology, Department of Natural Sciences, Novosibirsk State University, 630090 Novosibirsk, Russia; netesov.s@nsu.ru; 3Department of Medical Genetics and Postgenomic Technologies, Federal State Autonomous Educational Institution of Higher Education I M Sechenov First Moscow State Medical University of the Ministry of Health of the Russian Federation (Sechenov University), 119991 Moscow, Russia; 4Department of Virology, Faculty of Biology, Lomonosov Moscow State University, 119234 Moscow, Russia; 5National Medical Research Center of Phthisiopulmonology and Infectious Diseases, 127473 Moscow, Russia; vladimir@chulanov.ru; 6Infectious Diseases Department, Federal State Autonomous Educational Institution of Higher Education I M Sechenov First Moscow State Medical University of the Ministry of Health of the Russian Federation (Sechenov University), 119991 Moscow, Russia; 7Center for Precision Genetic Technologies for Medicine, Engelhardt Institute of Molecular Biology, Russian Academy of Sciences, 119991 Moscow, Russia; 8Laboratory of Molecular Epidemiology of Viral Hepatitis, Central Research Institute of Epidemiology, 111123 Moscow, Russia; isaeva.06@mail.ru (O.V.I.); karen-kyuregyan@yandex.ru (K.K.K.); michmich2@yandex.ru (M.I.M.); 9Laboratory of Viral Hepatitis, Mechnikov Research Institute of Vaccines and Sera, 105064 Moscow, Russia; 10Russian Medical Academy of Continuous Professional Education, 125993 Moscow, Russia; 11Department of Infectiology and Virology, Federal State Autonomous Educational Institution of Higher Education I M Sechenov First Moscow State Medical University of the Ministry of Health of the Russian Federation (Sechenov University), 119991 Moscow, Russia

**Keywords:** hepatitis C virus, anti-HCV, HCV RNA, Russian Federation, conditionally healthy population, HCV prevalence

## Abstract

Identifying population groups at greatest risk of hepatitis C virus (HCV) infection is essential for targeting screening and treatment. We analyzed the seroprevalence of antibodies to HCV (anti-HCV) and HCV RNA in serum samples from 37,291 conditionally healthy volunteers collected between 2018 and 2022, and from 4764 individuals sampled in 2008, totaling 42,055 participants. In 2018–2022, anti-HCV prevalence varied by region, ranging from 1.1 to 1.4% in Belgorod, Moscow, and St. Petersburg to 1.8–2.1% in Dagestan, Tatarstan, Novosibirsk, Tyva, and southern Yakutia, and reaching 3.4–5.2% in Khabarovsk and the Arctic zone of Yakutia. In 2008, prevalence in Moscow, Rostov, Sverdlovsk, Tyva, and Yakutia ranged from 1.7% to 3.3%. A significant decline over time was observed: from a mean of 2.6 ± 0.5% in 2008 to 1.9 ± 0.1% in 2018–2022 (*p* < 0.01). In recent years, men were more frequently anti-HCV-positive than women (2.5 ± 0.2% vs. 1.5 ± 0.2%, *p* < 0.01), whereas no sex differences were noted in 2008. The age of a sharp prevalence increase shifted from the 20–29 cohort in 2008 to the 30–39 cohort in 2018–2022. Based on the demographic structure, we estimate ~3.23 million anti-HCV carriers in Russia. HCV RNA was detectable in only one-third of seropositive individuals, identifying them as candidates for antiviral therapy. Thus, in contemporary Russia, men aged over 30 years have the highest risk of HCV infection and should be prioritized for targeted screening.

## 1. Introduction

Direct-acting antiviral (DAA) therapies can cure up to 95% of patients with hepatitis C virus (HCV) infection [1]. This makes the identification and timely treatment of infected individuals a cornerstone strategy not only to reduce mortality from cirrhosis and hepatocellular carcinoma, but also to interrupt onward transmission [2].

Chronic hepatitis C (CHC) is often asymptomatic, with up to 90% of infected individuals unaware of their status [3]. Many of them do not seek medical care, yet remain a source of potential transmission. Such individuals are typically identified incidentally, e.g., during medical visits for unrelated reasons, or through targeted screening of selected subpopulations within the conditionally healthy population.

Population-wide screening would be the most appropriate approach; however, it has never been conducted in Russia, likely due to administrative challenges and high costs. A practical alternative is to focus on identifying subgroups (defined by region, sex, age, or social characteristics) where the proportion of HCV carriers is highest. Data from such studies can be used for the development of targeted diagnostic and treatment programs.

The primary serological marker of HCV infection is the presence of antibodies to HCV (anti-HCV), representing both IgG and IgM against viral antigens [4]. Detection of HCV RNA by PCR serves two purposes: confirming chronic infection—an indication for initiating antiviral therapy according to clinical guidelines [5], and identifying active viral replication. Individuals with detectable HCV RNA derive the greatest benefit from treatment and should be prioritized for it [6].

HCV remains a significant public health problem in the Russian Federation. National surveillance data show that, over the past decade (2014–2024), the incidence of acute hepatitis C (AHC) has remained stable at 1–1.5 cases per 100,000 annually, while CHC has been reported at 31–40 cases per 100,000 [7,8]. Long-term incidence trends show no consistent decline (Figure 1); the transient drop during 2020–2021 likely reflects COVID-19–related disruptions to healthcare [7].

Information on the main risk factors of HCV infection in modern Russia remains quite inconsistent, likely reflecting the absence of a single dominant risk factor. Authors of review papers on this topic (e.g., [9,10,11]) generally agree that most individuals currently living with CHC in Russia acquired the virus between the late 1990s and around 2010, when intravenous drug use was the principal and clearly predominant mode of the HCV transmission [12,13]. These days, however, this factor no longer plays such a decisive role (though it remains important), having been surpassed by medical procedures, including blood transfusions [9,10,11]. Other commonly cited risk factors include perinatal transmission, tattoos and piercings, imprisonment, and, notably, sexual contact with infected partners among young people [9]. The distribution and relative impact of these factors, however, appear to vary considerably across regions and even among different social groups within the same region [9,10,11].

Some estimates suggest that ~1.8% of the Russian population are HCV carriers [14], that is a moderately high rate globally. For comparison, prevalence is estimated at 0.1–0.2% in Western Europe, 0.3–0.4% in China and India, and 0.7% in the United States; higher rates are reported in Ukraine (3.3%) and Pakistan (3.8%) [14]. Given its population size, Russia ranks among the top five countries worldwide in absolute number of carriers, estimated at 2.7 million [14]. Another estimates, based on the mathematical modeling [15] suggests even higher numbers, up to 4.25 million (2.9%) in 2020.

National averages, however, mask substantial heterogeneity across Russia’s diverse geographic, economic, and demographic regions. Studies in conditionally healthy populations have reported anti-HCV prevalence both far below and well above the national average. Within the same Sakha Republic (Yakutia), for example, estimates have ranged from 0–1% to 10–13%, depending on climatic zone (Arctic or southern districts) and study period (1999–2002 vs. 2005–2006) [16,17].

In this context, we aimed to assess age- and sex-specific HCV prevalence in a conditionally healthy population across twelve Russian regions (Table 1), drawing on both original data and published studies. Our findings are intended to help public health authorities better identify high-risk groups for targeted screening and to ensure timely access to antiviral therapy.

## 2. Materials and Methods

### 2.1. Study Design and Samples Collection

A total of 42,055 healthy volunteers from twelve regions across the Russian Federation, spanning from west to east, participated in the serosurvey conducted over multiple years (Table 1, Figure 2). Data from 30,724 of these participants are presented here for the first time. The data obtained from an additional 11,331 volunteers were previously published, primarily in Russian-language sources (see references in Table 1). To enable comparison with earlier findings, we also refer to previous studies by other authors, particularly those conducted in the 1990s and early 2000s (see Section 3.2 for details).

In the main text, as well as in the tables and figure captions, we use abbreviated names for the studied regions. Specifically, Kaliningrad, Belgorod, Rostov, Sverdlovsk, Novosibirsk, and Khabarovsk regions are referred to by the names of their capital cities. For the republics, we use the names Dagestan, Tatarstan, and Tyva. “Moscow” and “St. Petersburg” refer to the combined populations of Moscow city with Moscow Region, and St. Petersburg with Leningrad Region, respectively. In the Republic of Sakha (Yakutia), we identified two distinct groups. The group generally referred to as “Yakutia” represents the population of the so-called “agricultural zone” (southern districts, including the major cities of Yakutsk and Neryungri), studied in 2008 and 2018 (Table 1). The “Arctic zone of Yakutia” refers to the northern Momsky District, surveyed in 2022 (Table 1). This distinction is important, as the southern and northern regions of Yakutia have shown marked differences in the epidemiology and prevalence of parenteral hepatitis [16,17].

The study included male and female participants aged 0 to 95 years, divided into nine age groups: under 1 year, 1–9, 10–14, 15–19, 20–29, 30–39, 40–49, 50–59, and ≥60 years. The number of participants in each age group and region is shown in Appendix A. The male-to-female ratio ranged from 1:1 to 1:2.1, depending on the region (Table 1 and Appendix A). Ages were grouped into relatively broad 10-year intervals to ensure that each group contained a sufficient number of participants for statistical reliability. However, within adolescence (10–20 years), two five-year subgroups were identified, as this age range was previously associated with HCV infection linked to drug use (particularly in the late 1990s and 2000s; see [12,13] for details).

All participants met the following inclusion criteria: apparently healthy, with no obvious symptoms of acute illness at the time of enrollment, and permanent residency in the study region. Exclusion criteria included a history of liver disease (infectious or non-infectious), any current acute illness/body temperature above 37.1 °C; as well as any surgery, blood transfusion, or treatment with blood products in the three months prior to enrollment (self-reported or, for participants under 15 years of age, reported by a parent or guardian).

All serum samples were coded, aliquoted, and stored at −70 °C until testing.

### 2.2. Ethics

The study was conducted in accordance with the principles of the World Medical Association Declaration of Helsinki for ethical medical research involving human subjects. Informed written consent was signed by all the participants or their legal guardians. The study protocol was approved by the Independent Interdisciplinary Ethics Committee for the Ethical Review of Clinical Research, Moscow, Russia (Approval No. 17, 16 November 2019), and by the Ethics Committee of the Mechnikov Research Institute for Vaccines and Sera, Moscow, Russia (Approval No. 1, 28 February 2018). The study conducted in 2008 was approved by the Ethics Committee of the Chumakov Institute of Poliomyelitis and Viral Encephalitis, Moscow, Russia (Approval No. 6, 1 April 2008).

### 2.3. ELISA Testing

Serum samples from Belgorod region (Table 1) were tested for anti-HCV antibodies using the Architect Anti-HCV test (Abbott Laboratories, Abbott Park, IL, USA). Samples reactive in the screening test were confirmed by immunoblotting for antibodies to structural and non-structural HCV proteins (INNO-LIA HCV, Fujirebio Europe N.V., Gent, Belgium). All other serum samples listed in Table 1 were tested for anti-HCV (IgG + IgM) using commercial ELISA kits (Vector-Best, Novosibirsk, Russia). Samples that tested positive in the screening assay were subsequently analyzed using the anti-HCV confirmation test from the same manufacturer. A participant was classified as an anti-HCV-positive carrier if their sample tested positive in the confirmation ELISA.

### 2.4. PCR Testing

All samples that were found to be positive in the anti-HCV confirmation assay, were further examined using a commercial real-time PCR kit for qualitative detection of HCV RNA (AmpliSens, Moscow, Russia). This assay has a detection limit of 10 IU/mL when nucleic acids are extracted from a 1 mL sample. Samples collected in Moscow, St. Petersburg, Dagestan, Novosibirsk, and Khabarovsk (2018–2020) that were negative for anti-HCV by ELISA were also tested by PCR in pools of 10 samples using the same AmpliSens reagent kit. If a pooled sample tested positive for HCV RNA, each of the 10 individual samples in the pool was subsequently tested to identify the HCV RNA-positive sample.

### 2.5. Statistical Analysis

Statistical analysis included assessment of differences in proportions of anti-HCV carriers in groups/cohorts using the Chi-square test with Yates’ correction, or Fisher’s exact test for small samples. A *p*-value < 0.05 was considered statistically significant. Confidence intervals (95% CI) for proportions were calculated using the binomial distribution. Correlation coefficients (r) between quantitative variables, such as the incidence of hepatitis C and anti-HCV prevalence, were determined using the two-tailed Spearman’s non-parametric test in Prism software, version 9.0.2 (GraphPad, Boston, MA, USA).

## 3. Results

### 3.1. Differences in Anti-HCV Prevalence Among Regional Groups

The prevalence of antibodies to the hepatitis C virus (anti-HCV prevalence) in the regional groups surveyed in 2018–2022 is summarized in Figure 2 and Table 2 (with separate data for men and women). Detailed age stratification data are provided in Appendix A. Table 3 presents the results of pairwise comparisons between regional groups from 2018 to 2022, based on the proportions of anti-HCV-positive individuals, using the chi-square test.

Based on these results, we considered it possible to broadly classify regional groups that were observed in 2018–2022 into three categories. The regions with a relatively low prevalence of anti-HCV (1.1–1.4%) included Belgorod Region, Moscow (meaning Moscow Region along with Moscow City), and Saint Petersburg (including Saint Petersburg City and Leningrad Region). Differences among these regions were not statistically significant (*p* > 0.05).

“Moderate” anti-HCV prevalence (1.8–2.1%) was found in Republic of Dagestan, Republic of Tatarstan, Novosibirsk Region, Republic of Tyva, and Republic of Sakha (Yakutia, southern districts). Differences within this group were also not statistically significant (*p* > 0.05).

Finally, relatively high prevalence (3.4–5.2%) was reported for Khabarovsk Region and Yakutia (Arctic zone). Differences between these two regions were not statistically significant (*p* > 0.05; Table 3).

Kaliningrad Region occupied an intermediate position, statistically similar to both regions with the “moderate” prevalence (Republic of Tatarstan and Republic of Tyva) and to the “high-prevalence” region of Khabarovsk. Notably, two “moderate-prevalence” regions (Republic of Tatarstan and Republic of Tyva) did not differ significantly from “low-prevalence” regions (Saint Petersburg and Moscow). Apart from these overlaps, differences between the three prevalence categories were statistically significant (Table 3).

Data for the 2008 survey are also presented in Figure 2, Table 2 and Appendix A. In 2008, both low- and high-prevalence groups were identifiable: Moscow (1.7%) represented the lower end, while the Republic of Tyva and Yakutia (3.3%), were at the higher end (*p* < 0.05, Table 2). Rostov Region and Sverdlovsk Region occupied intermediate positions (2.1–3.0%) and did not differ significantly from any of the other regions in 2008.

These findings indicate that, at least up to the recent 2018–2022 period, the prevalence of HCV infection in the Russian Federation has remained markedly heterogeneous across regions.

### 3.2. Dynamics of Anti-HCV Prevalence in Regional Groups over Time

The anti-HCV prevalence in the conditionally healthy population across the surveyed regions at two time points (2008 and 2018–2022; see Table 2 and Table 3) made it possible to evaluate temporal trends. Additional conclusions were drawn by comparing our findings with similar studies performed by other researchers in the same regions. All the data for available time points is presented in Figure 2.

In Moscow, the proportion of anti-HCV carriers decreased from 1.7% in 2008 to 1.2% in 2018–2019. However, this decrease was not statistically significant (see confidence intervals in Table 2).

For Novosibirsk region, data on the anti-HCV prevalence in the conditionally healthy population were previously reported for the period 1995–1999 [25]. In that study, a group of 1073 individuals were examined, including schoolchildren, medical students, and randomly selected adults, with a mean anti-HCV prevalence of 4.2 ± 1.2%. Another study [12], conducted in 2000–2002, reported a prevalence of 5.6 ± 2.1% among 500 patients attending a non-infectious outpatient clinic. The prevalence estimates from these two earlier studies did not differ significantly from each other, but both were significantly higher than the value obtained in our study for Novosibirsk Region in 2019–2020 (1.9 ± 0.3%, *p*-value < 0.01; Table 2).

In the Republic of Tyva, the proportion of anti-HCV-positive carriers also declined from 3.3% in 2008 to 2.0% in 2019; however, this difference was not statistically significant (*p*-value > 0.05).

The proportion of anti-HCV carriers among the conditionally healthy population of Yakutia (excluding the Arctic zone; see below), as observed in our study in 2018, was 2.0%, which was lower than the 3.3% recorded in 2008 (Table 2), although the difference was not statistically significant. At the same time, a comparison with earlier results reported from the same southern regions of Yakutia reveals a more pronounced downward trend. In particular, a study conducted between 1999 and 2002 [16], which included 1394 adolescents and adults from the southern “agricultural” districts of Yakutia (Namsky, Gorny, Vilyuisky), reported a high mean anti-HCV prevalence of 6.2 ± 1.3% (ranging from 2.7% to 13.4% depending on age). A subsequent study carried out in 2005–2006 in the city of Neryungri, also located in southern Yakutia, examined 329 conditionally healthy individuals and reported an even higher prevalence of 13.1 ± 3.9% [17]. Both of these values were significantly higher than those observed in our study for the same “agricultural” zone in 2008 (3.3%) and, especially, in 2018 (2.0%) (Table 2; *p*-value < 0.01 for all comparisons).

Thus, the available data suggest a decreasing trend in the prevalence of antibodies to the HCV in these regions from 2008 (or earlier) to more recent years (2018–2020), although this trend was not always statistically significant.

However, this decline was not observed in all study groups. For example, in the same study cited above [16], a very low prevalence of antibodies to the HCV was recorded in the Abyisky and Eveno-Byntaisky districts of the Arctic zone of Yakutia: 0% among 296 children and 162 adolescents, and 1.1% among 370 adults. It should be noted that these figures refer to the northern regions of Yakutia, in contrast to the southern areas discussed above.

This low prevalence was confirmed by another study conducted in 2005–2006 [17], which reported the anti-HCV prevalence of only 1.3 ± 1.3% among 301 representatives of the Evenki ethnic group, who predominantly inhabit the northern territories of Yakutia. In contrast, our 2022 study revealed a markedly higher prevalence in residents of the Momsky district, also located in the Arctic zone and bordering the Abyisky district, with anti-HCV detected in 5.2 ± 2.3% of the examined individuals, that was the highest rate among all groups included in Table 2. The reasons for this sharp increase remain unclear and require further investigation.

Similarly, the increase in anti-HCV prevalence over time was observed in the Republic of Dagestan (North Caucasus). In 2019–2020 our study found the anti-HCV prevalence of 1.8% in this region (Table 2), whereas an earlier investigation of 10,682 blood donor samples collected in 1994–1996 reported a significantly lower rate of 0.9 ± 0.2% (*p*-value < 0.01) [26]. One explanation for this difference may be that during the earlier period, the Republic of Dagestan had probably not yet experienced the surge in hepatitis C incidence that peaked in the Russian Federation in the late 1990s [12,13]. Additionally, donor cohorts (such as a studied in the cited investigation) typically have lower morbidity from parenterally transmitted infections than the general population [13].

Overall, if we assume that the volunteer groups examined in our study are at least partially representative of the conditionally healthy population of the Russian Federation (taking into account the limitations discussed further in the Section 4) it can be concluded that the proportion of anti-HCV carriers in the country has significantly declined over the past 10–12 years. In the combined group of volunteers studied in 2008 (4764 individuals from five regions differing in economic, geographic, and climatic characteristics), the mean prevalence of anti-HCV was 2.6 ± 0.5% (see bottom lines of Table 2). By 2018–2022 (37,291 participants from ten regions), this average prevalence had significantly decreased to 1.9 ± 0.1% (Table 2, *p*-value < 0.01). However, as illustrated by the examples of rising anti-HCV prevalence in some regions, it cannot yet be concluded that the overall downward trend is consistent or stable across the entire country. This uncertainty is also reflected in the slow and gradual decline in the official number of newly registered cases of hepatitis C over the years (Figure 1), that is discussed in the following section.

### 3.3. Correlation Between Hepatitis C Incidence and Anti-HCV Prevalence in Regional Groups

The pronounced regional heterogeneity described above is also reflected in the officially reported incidence rates of hepatitis C. Figure 1 illustrates the dynamics of long-term cumulative incidence, including both acute hepatitis C and chronic hepatitis C, while Appendix A provides detailed data on the incidence in the surveyed regions for the year in which the samples were collected, presented separately for acute hepatitis C and chronic hepatitis C.

The data show that hepatitis C incidence rates do not always correspond to the prevalence of anti-HCV. For example, Saint Petersburg has consistently recorded the highest incidence of hepatitis C in the Russian Federation, 70 to 90 cases per 100,000 population annually (including both acute hepatitis C and chronic hepatitis C). However, the prevalence of anti-HCV in this region was among the lowest, at only 1.4% (Table 2). Similarly high incidence rates were reported in Novosibirsk Region (50–90 per 100,000 annually), despite a moderate anti-HCV prevalence of 1.9%. In Moscow, where the prevalence of anti-HCV was low (1.2%), the long-term incidence of hepatitis C remained at 35–55 cases per 100,000 annually, exceeding the national average of 30–40 per 100,000 (Figure 1).

In contrast, in the Republic of Dagestan, the incidence of hepatitis C has remained below 10 per 100,000 annually for more than a decade, yet the prevalence of anti-HCV was 1.8%—comparable to that of Novosibirsk Region, despite much higher incidence rate there.

Formal correlation analysis using the two-tailed Spearman rank test revealed no statistically significant association between anti-HCV prevalence and the regional incidence of either acute or chronic hepatitis C in the year of sample collection, or with their combined incidence. In all cases, correlation coefficients (r) ranged from –0.12 to –0.01, with corresponding *p*-values between 0.63 and 0.95 (see Appendix A).

### 3.4. Dynamics of Anti-HCV Prevalence in Age Groups

Appendix A presents data on the anti-HCV prevalence among participants in the following age categories: under 1 year, 1–9 years, 10–14 years, 15–19 years, 20–29 years, 30–39 years, 40–49 years, 50–59 years, and 60 years or older, across the surveyed regions. Based on these data, age-dependent prevalence curves were constructed and are shown in Figure 3 (A: groups surveyed in 2018–2022; B: groups surveyed in 2008).

The graphs clearly demonstrate that anti-HCV prevalence increases with age. In the younger age groups, the proportion of positive carriers remains consistently low. Among participants aged 1 to 19 years, anti-HCV prevalence in almost all regions remained below 1%. Minor deviations from this trend, likely representing random fluctuations, were observed in the 1–9-year age group in Sverdlovsk Region (2008) and in the 10–14-year age group in Kaliningrad Region (2019) (Figure 3).

A marked increase in anti-HCV prevalence begins in adulthood. In the 2008 cohort, this increase occurred starting from the 20–29-year age group, whereas in the 2018–2022 cohort it was delayed until the 30–39-year age group. This shift is evident not only visually in the graphs but is also supported by statistical analysis. For example, in the entire 2008 group (depicted as the thick black line in Figure 3B), the differences in prevalence between adjacent younger age groups (for example, <1 year vs. 1–9 years, 1–9 years vs. 10–14 years, 10–14 years vs. 15–19 years) were not statistically significant. However, a significant difference was observed between the 15–19-year age group and the 20–29-year age group (1.2 ± 0.9% vs. 3.1 ± 1.5%, *p*-value < 0.05; Appendix A).

In contrast, for the 2018–2022 cohort, the first statistically significant increase occurred between the 20–29-year age group and the 30–39-year age group (0.8 ± 0.2% vs. 2.0 ± 0.3%, *p*-value < 0.01; Appendix A). No significant increases were observed in age groups younger than 20 years, with the exception of the <1-year group, which is discussed additionally below. Another statistically significant rise was observed after the age of 39 years: in the 40–49-year age group, anti-HCV prevalence reached 3.6 ± 0.6%, compared with 2.0 ± 0.3% in the 30–39-year age group (*p*-value < 0.01; Appendix A). Beyond the age of 49 years, changes in prevalence occurred more gradually (Figure 3A).

As noted previously, in several regions surveyed in 2018–2022—including Kaliningrad Region, Moscow, Tatarstan, Yakutia (non-Arctic zone), and Khabarovsk—an unusually high proportion of anti-HCV carriers was identified among children younger than one year (Figure 3 and Table 2). Moreover, in the entire 2018–2022 cohort, the prevalence of anti-HCV among infants younger than one year was 1.5 ± 0.8% (Table 2, line 17), which was significantly higher than the prevalence among children aged 1–9 years (0.8 ± 0.2%; *p*-value < 0.05). Since no reports of hepatitis C outbreaks specifically affecting newborns in the Russian Federation were identified, we believe that this elevated prevalence of antibodies in infants is most likely the result of passive transfer of maternal antibodies from anti-HCV-positive mothers. Supporting this interpretation, the prevalence of anti-HCV among infants in the regions mentioned (ranging from 1.9% to 2.9%, and reaching as high as 4.9% in Tatarstan) closely mirrored the prevalence observed among women of childbearing age (30–39 years) in the same regions (0.7–3.7%; Appendix A).

The absence of a similar peak in the <1-year age group in the 2008 data may be explained by the fact that, at that time, the number of women infected with the HCV (or carrying anti-HCV antibodies) who had recently given birth was likely too low for their infants to be represented in the study sample. In any case, based on the trends illustrated in Figure 3, maternal antibodies appear to persist in newborns for approximately one year.

### 3.5. Anti-HCV Prevalence in Men and Women

Table 2 presents the anti-HCV prevalence separately for men and women in each surveyed regional group. More detailed information, including the number of male and female anti-HCV carriers in specific age groups, is provided in Appendix A.

In all regional groups surveyed in 2008, no statistically significant differences in anti-HCV prevalence between men and women were observed (Table 2). On average, in the combined 2008 cohort, anti-HCV prevalence was 2.6 ± 0.7% among men and 2.7 ± 0.6% among women.

In contrast, in many of the regional groups surveyed during 2018–2022, the prevalence of anti-HCV was noticeably higher among men compared with women. Statistically significant differences were observed in the following groups: Saint Petersburg (2.4% versus 0.9%, *p*-value < 0.01), Belgorod (1.6% versus 0.7%, *p*-value < 0.05), Moscow (2.2% versus 0.9%, *p*-value < 0.01), Dagestan (2.2% versus 1.3%, *p*-value < 0.05), and Khabarovsk (4.2% versus 2.4%, *p*-value < 0.01). In the remaining regions, differences between male and female cohorts were not statistically significant (Table 2).

In the entire 2018–2022 cohort, anti-HCV prevalence among men (2.5 ± 0.2%) was significantly higher than among women (1.5 ± 0.2%, *p*-value < 0.01; Table 2). Although the identification of gender-specific risk factors for anti-HCV carriage was beyond the scope of this study, the observed data support the conclusion that, in contemporary Russian society, men are at a higher risk of infection with the HCV compared with women.

### 3.6. Results of PCR Testing

Data on participants who tested positive for HCV RNA by PCR are presented in Table 2. In the vast majority of cases, individuals who were PCR-positive for HCV RNA were also positive for antibodies to the HCV. That was expected, since PCR testing in this study was primarily conducted for samples that were seropositive in anti-HCV screening and confirmation assays.

However, in a number of large regional groups surveyed between 2018 and 2020—specifically Moscow, Saint Petersburg, Dagestan, Novosibirsk, and Khabarovsk—PCR testing was also performed on anti-HCV-negative samples using pooled sample analysis (ten samples per pool). In total, 30,232 anti-HCV-negative samples were tested in this manner, which resulted in the detection of 30 additional HCV RNA-positive sera (0.1%). These RNA-positive individuals are not distinguished in Table 2 from those identified through PCR testing of anti-HCV-positive samples.

Notably, all but one of these “seronegative” HCV RNA-positive samples were positive for antibodies to the hepatitis B core antigen (anti-HBcAg), a serological marker of chronic or past hepatitis B virus infection, and lacked any other serological markers of parenterally transmitted hepatitis viruses (that is, they were negative for hepatitis B surface antigen—HBsAg, negative for antibodies to hepatitis B surface antigen—anti-HBs IgG, and negative for antibodies to the HCV). Further details on the serological profiles of the studied groups with respect to hepatitis B virus infection can be found in our previous publication [27].

Among individuals who were positive for anti-HCV, the proportion of those who were also positive for HCV RNA ranged from 20% to 62% across the different regional groups (Table 2). When comparing the aggregated data from the 2008 and 2018–2022 cohorts, this proportion was similar: 38.5% and 32.6%, respectively. The relatively high number of individuals who were anti-HCV-positive but HCV RNA-negative is most likely explained by spontaneous clearance of the virus, which occurs in approximately 20–40% of individuals following an acute HCV infection [28].

### 3.7. Estimation of the Present Number of Individuals Carrying Anti-HCV in the Russian Federation

Attempts to estimate the number of individuals carrying markers of infectious diseases at the national level based on limited sample data are, by nature, speculative. The reasons for this are discussed in detail in the “Limitations of the Study” subsection at the end of the Discussion. Nevertheless, such estimates remain a favorite exercise among epidemiologists and can be valuable, particularly for planning the economic resources required for identifying and treating infected individuals. In this context, we present our own estimate, while acknowledging its inherent limitations.

In the combined group studied during 2018–2022 (37,291 participants; see Table 2 and Table 3), the overall prevalence of antibodies to HCV was found to be 1.9 ± 0.1%. However, as discussed in Section 3.4 and Section 3.5, anti-HCV prevalence varies significantly depending on age and sex. Furthermore, Appendix A shows that the age and sex distribution of our surveyed groups differed considerably from that of the general population.

For instance, individuals aged 30–39 years constituted 18% of the actual population in the Moscow region (including the city of Moscow) in 2019 [29] (the population data from the last pre-COVID-19 year will be used as a base in this section) but accounted for 36% of our surveyed group in 2018–2019 (Appendix A). In Dagestan, children aged 1–9 years represented 23% of the sample, compared to only 15% of the republic’s actual population. The overall male-to-female ratio in the 2018–2022 sample was 1:1.3 (Table 2), whereas the national ratio in 2019 was 1:1.15 [29], with similar discrepancies observed in other regions. Therefore, it was necessary to adjust the experimental sample to better reflect the true demographic structure of the Russian population.

To achieve this, we applied the age- and sex-specific prevalence rates of anti-HCV observed in our experimental sample to the actual demographic structure of each region. In practical terms, if the observed prevalence of anti-HCV in a given sex–age group was q%, and the size of that demographic stratum in the real population of a region was P, then the estimated number of anti-HCV carriers in that group would be P × q%. Summing the results across all sex–age strata yielded the total estimated number of anti-HCV carriers in the region (see Appendix A for detailed calculations).

This approach, however, does not allow the calculation of meaningful aggregated confidence intervals, as their combination would produce an excessively wide range. Additionally, data from the Momsky District of Yakutia were excluded from this calculation due to its small population size.

Applying this method to all regions included in the 2019–2022 study—covering a total population of 42 million in 2019—produced an estimate of approximately 937,000 individuals carrying antibodies to HCV, corresponding to a prevalence of 2.2%. This value is slightly higher than the prevalence observed directly in the experimental sample (1.9 ± 0.1%; 713 carriers among 37,291 participants; see Table 2). We consider this higher estimate more appropriate for the purposes of planning national HCV control strategies, given the methodological validity of the demographic adjustment.

If we further assume that this 2.2% prevalence applies to the entire population of the Russian Federation (approximately 146 million in 2019), the estimated total number of individuals carrying anti-HCV nationwide would be about 3.23 million. This figure aligns with previously published estimates ranging from 2.7 to 4.5 million [14,15]. Of course, our estimate is based on a relatively small sample compared with the total population of Russia, and therefore may differ from results obtained in larger-scale studies. The Appendix A provides the complete experimental dataset and corresponding demographic statistics, enabling other researchers to perform more refined calculations should new or additional information become available in the future.

## 4. Discussion

In this study, we investigated the prevalence of antibodies to the HCV by direct In this study, we examined the prevalence of antibodies to HCV through direct serological testing of blood samples from volunteers representing various population groups. Between 2018 and 2022, a total of 37,291 unique samples were collected from the conditionally healthy population of all ages across ten geographical regions of the Russian Federation. Importantly, approximately 82.5% of these samples were collected before the end of 2019, i.e., before the onset of the COVID-19 pandemic, which otherwise would have delayed this epidemiological study. Additionally, 4764 samples collected in 2008 were analyzed. Based on this dataset, the average prevalence of anti-HCV was estimated at 1.9 ± 0.1% for the 2018–2022 cohort and 2.8 ± 0.5% for the 2008 cohort (Table 2).

Notable regional differences in anti-HCV prevalence were found. In the 2018–2022 cohort, the lowest prevalence values were recorded in the Belgorod region, Moscow, and Saint Petersburg (1.1–1.4%), while the highest values were recorded in Khabarovsk region and the Arctic zone of Yakutia (3.4–5.2%). These differences were statistically significant (Table 3).

Overall, the results indicate a general downward trend in HCV prevalence over time. However, this trend is not universal: in the Dagestan and the Arctic zone of Yakutia, the proportion of anti-HCV-positive individuals in 2018–2020 exceeded the levels reported in earlier studies by other authors.

An important finding was the absence of correlation between anti-HCV prevalence and the officially registered incidence of hepatitis C (see Section 3.3). This suggests that official incidence rates are not reliable indicators of the true morbidity of infection in the population but instead reflect the extent of diagnostic testing coverage in different regions.

Substantial differences were also observed in the prevalence of anti-HCV across age groups. In most younger age groups examined in 2018–2022, the prevalence did not exceed 1%. However, starting from approximately 30 years of age, prevalence increased markedly to 2–4%, and in older age groups, it reached 4–10% (Figure 3A, Appendix A). In the 2008 cohort, this increase was already evident starting from the 20–29-year age group. It is plausible that individuals in this cohort were infected during adolescence or early adulthood in the late 1990s and early 2000s, when predominant risk factors (especially injection drug use) were more widespread [12,13]. Children and adolescents under 20 years of age examined in 2008, by contrast, likely had significantly lower lifetime exposure to these risk factors.

In the 2018–2022 cohort, the age-related increase in anti-HCV prevalence has shifted by approximately one decade compared to the 2008 data. This suggests that the same cohort of individuals likely infected in the late 1990s to early 2000s had aged by ten years and were now represented in older age groups. Meanwhile, individuals under 30 years of age in 2018–2022 continued to show relatively low infection rates, suggesting that high-risk factors prevalent among young people in the 1990s are no longer as influential for the younger generation. This shift may reflect positive changes in public health and behavioral patterns.

At that, in contemporary Russian society, certain as yet unidentified risk factors appear to be disproportionately associated with the male sex. In the entire 2018–2022 cohort, the prevalence of anti-HCV among men was significantly higher than among women (2.5% versus 1.5%; see Section 3.5). No such sex-related difference was detected in the 2008 cohort, suggesting that before 2008, there were no risk factors disproportionately affecting one sex.

Taken together, these findings underscore the pronounced heterogeneity in the HCV epidemiology across the Russian Federation. Seroepidemiological studies such as ours help to identify demographic and regional groups most affected by the infection, which should be prioritized in prevention and control programs. Based on the age- and sex-specific patterns identified, it can be concluded that in modern Russia (2018–2022), the majority of infected individuals are men over 30 years of age (Figure 3A, Table 2). Detailed numerical data are provided in Appendix A, which can be regarded not only as reference material but also as a basis for designing targeted public health interventions.

In accordance with the aims outlined in the Introduction, diagnostic screening followed by antiviral treatment would be most effective if directed at groups with the highest concentration of seropositive individuals. Adjusting for the contribution of different age and sex cohorts in the study sample, we estimated that the total number of anti-HCV carriers in modern Russia is approximately 3.23 million individuals. As demonstrated in our study, only about one-third of individuals testing positive for anti-HCV also test positive for the HCV RNA and thus have a clinical indication for antiviral therapy [5,6].

The remaining anti-HCV-positive but RNA-negative individuals require only observation. According to current Russian guidelines, such individuals undergo two polymerase chain reaction tests six months apart, and if both results are negative, they are considered recovered and removed from the viral hepatitis register [30].

Based on our estimates, approximately 1.1 million people in the Russian Federation require antiviral treatment. At an average cost of 200,000–300,000 rubles per course (equivalent to USD 2000–3000), the total projected cost of therapy would amount to 200–300 billion rubles (USD 2–3 billion). For comparison, official estimates place the annual economic burden of chronic hepatitis C in Russia at 65–75 billion rubles [7,8]. Therefore, the economic benefits of implementing large-scale screening and treatment programs could become apparent within five to ten years. A significant step in this direction is the inclusion of anti-HCV screening in the adult preventive medical examination program beginning in 2024 [31] as well as the launch in 2024 of the federal project “Combating Hepatitis C” within the framework of the national program “Long and Active Life” [32], which includes a substantial expansion of access to antiviral treatment funded by the federal budget.

### Limitations of the Study

The present work—or more precisely, a set of studies conducted in different years but united by a common objective and presented here under a single title—has several methodological limitations. These limitations are characteristic of many seroepidemiological studies and are primarily associated with methodological simplifications that should be explicitly acknowledged:

1. Enrollment of participants in the study was based on inviting volunteers rather than employing a randomized sampling strategy that would statistically represent different social, economic, and behavioral strata of the population. Such an approach may have led to an underestimation of the anti-HCV prevalence in the surveyed groups. Volunteer-based recruitment usually does not adequately capture marginalized or high-risk groups, which, although relatively small in number, often have a disproportionately high prevalence of HCV infection [12]. Furthermore, the primary aim of the present work was to identify asymptomatic anti-HCV carriers “hidden” within the ostensibly healthy population. For this reason, individuals with a known diagnosis of HCV infection or with a history of other viral hepatitis or parenteral infections were deliberately excluded (self-reported or guards-reported hepatitis history served as the exclusion criterion for participants). Consequently, these individuals were not included in the statistical analyses of regional prevalence rates.

2. The regional cohorts presented in Table 1 and Table 2, although all derived from the “conditionally healthy” population, are not epidemiologically homogeneous with one another. While an effort was made to recruit a balanced distribution of participants across age and sex groups to ensure adequate statistical power for subgroup analysis, it was not feasible to perfectly match the demographic structure of the study samples to that of the actual population in each region. Moreover, it was not possible to refuse participation to volunteers who wished to be included, even if their demographic subgroup was already sufficiently represented. As a result, as shown in Table 1 and Appendix A, the representation of specific age and sex categories varied between regions (see also Section 3.7). This demographic imbalance should be considered when interpreting and comparing prevalence estimates across regions.

3. Almost no epidemiological data were collected alongside the samples (apart from demographic information), which prevents drawing reliable conclusions about the causes of observed disparities in anti-HCV prevalence among different groups. While we were able to describe the proportions of individuals carrying the marker within the population, we can only hypothesize about the epidemiological processes that have led to the current situation.

4. In this study, the primary analytical focus was on the prevalence of antibodies to the HCV, with the assumption that this marker correlates proportionally with the overall prevalence of HCV infection in the population. The results of PCR testing for HCV RNA were used only as information to indicate the proportion of “active”, that is, replicative, infections. As shown in Table 2, the proportion of anti-HCV-positive samples that were also positive for HCV RNA varied between 30% and 70% depending on the region. As discussed in Section 3.6, this discrepancy is most likely attributable to spontaneous clearance of the virus, which occurs in approximately 20–40% of individuals following acute infection [28]. However, it cannot be ruled out that some of the PCR-negative results could be explained by the possible hyper sensitivity of the ELISA assays used, or conversely, by the limited sensitivity of the PCR assays applied in certain cases. In any case, prevalence estimates based solely on anti-HCV seropositivity tend to overstate the proportion of individuals with active, chronic infection.

These limitations inevitably complicate the determination of the absolute number of individuals infected with HCV in the Russian Federation as a whole and in individual regions. Nevertheless, the consistent application of the same methodology across all study groups allows for the assessment of temporal trends within the same population (where multiple time points are available) and enables qualitative comparison between different regions. This approach has yielded several important and noteworthy findings, which are presented above and elaborated upon here.

5. One of the main limitations of this study is the absence of data on anti-HCV prevalence for the period between 2008 and 2018–2022. Therefore, the study does not provide a continuous 14-year trend but rather compares two distinct time points at the beginning and end of this interval. This gap may affect the interpretation of the observed trends: although a decline in anti-HCV prevalence is evident in many regions between 2008 and 2018–2022, it remains unclear whether this decrease was steady throughout the entire period or if fluctuations occurred in between.

## 5. Conclusions

In this large-scale seroepidemiological investigation, we examined the prevalence of antibodies to the HCV across diverse age, sex, and regional cohorts of the Russian Federation. The analysis was based on more than 42,000 serum samples collected over a fourteen-year period, from 2008 to 2022. This comprehensive dataset allowed for the assessment of both cross-sectional and temporal patterns in the HCV spread.

Our results demonstrated substantial variation in the prevalence of antibodies to the hepatitis C virus between different geographic regions and demographic groups. These findings confirm the pronounced heterogeneity of the HCV epidemiology within the Russian Federation. The data indicate a gradual overall decline in the prevalence of anti-HCV over time; however, this trend was not consistent across all surveyed regions, with some areas showing stable or even increasing prevalence rates.

By applying an adjustment for the actual age and sex distribution of the Russian population, we estimated that approximately 3.23 million individuals in the country are positive for antibodies to the HCV. Of these, roughly one-third are expected to have detectable HCV RNA and, therefore, meet clinical criteria for antiviral therapy. This proportion translates to an estimated 1.1 million individuals who require antiviral treatment.

These results highlight the importance of prioritizing targeted screening and treatment interventions for demographic groups with the highest prevalence rates. In particular, the findings indicate that men over 30 years of age constitute a key high-risk group in modern Russia. The integration of anti-HCV screening into the national program of preventive medical examinations for adults, along with the significant expansion of treatment programs beginning in 2024, represents a timely and strategically important step toward reducing the burden of HCV infection, improving population health outcomes, and ensuring a more efficient allocation of healthcare resources.

## Figures and Tables

**Figure 1 viruses-17-01529-f001:**
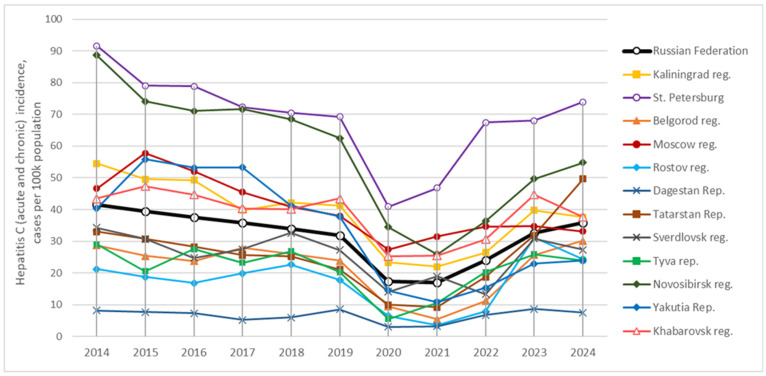
Cumulative incidence of acute and chronic hepatitis C in the studied regions, 2014–2024. Data are derived from official statistical reports [7,8] and regional health statistics provided by the Russian Ministry of Health.

**Figure 2 viruses-17-01529-f002:**
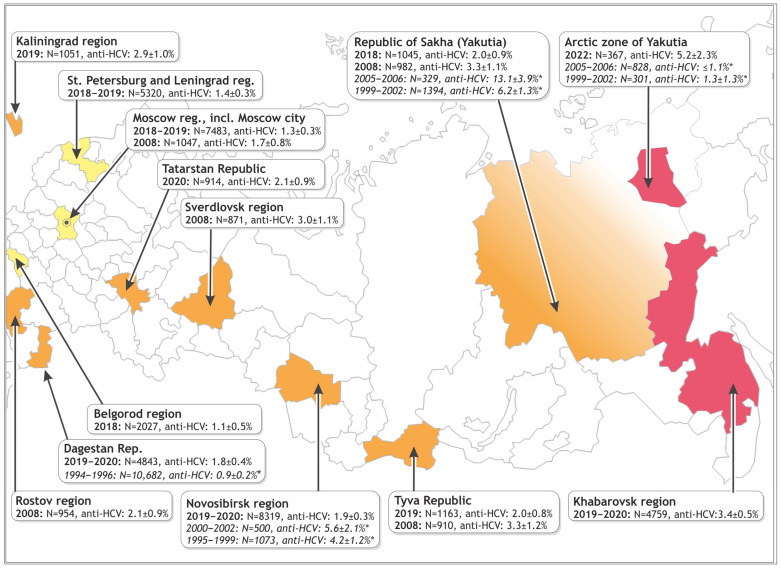
Map of sample collection sites with designated regions of the Russian Federation. The years of sample collection and the number of individuals studied are shown, along with the proportion of anti-HCV carriers (±95% CI). Data marked with an asterisk (*) were reported by other authors [12,16,17,25,26] and are included for comparison with our original results (see Section 3.2 for details).

**Figure 3 viruses-17-01529-f003:**
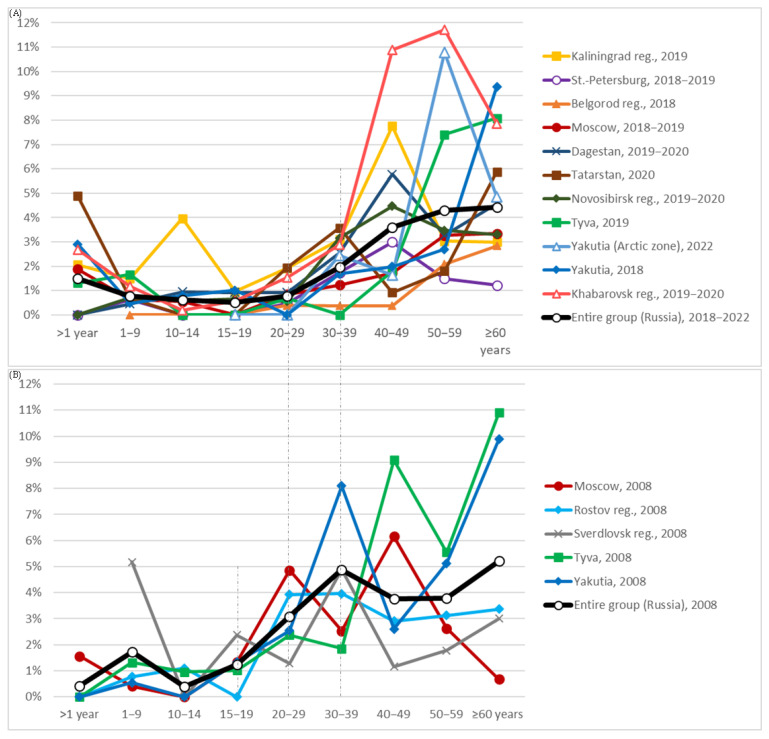
Prevalence of antibodies to the HCV (%) in age cohorts of the conditionally healthy population across the surveyed regions. (**A**)—data from regional groups examined in 2018–2022; (**B**)—data from groups examined in 2008. Numerical values correspond to those presented in Appendix A.

**Table 1 viruses-17-01529-t001:** Study groups (west to east).

Regional Group	Years of Collection	N	Proportion of Regional Population ^1^	Sex Ratio (M:F) ^2^	First Published in
Kaliningrad Region	2019	1051	0.11%	1:1.1	[18]
Saint Petersburg and Leningrad Region ^3^	2018–2019	5320	0.07%	1:2.2	This study
Belgorod Region	2018	2027	0.13%	1:1.2	[19]
Moscow Region ^3^ (including Moscow City)	2018–2019	7483	0.04%	1:2.1	This study
Moscow Region ^3^ (including Moscow City)	2008	1047	0.01%	1:1.4	[20]
Rostov Region	2008	954	0.02%	1:1.4	[20]
Republic of Dagestan	2019–2020	4843	0.15%	1:1.2	This study
Republic of Tatarstan	2020	914	0.02%	1:1.9	[21]
Sverdlovsk Region	2008	871	0.02%	1:1.4	[20]
Novosibirsk Region	2019–2020	8319	0.30%	1:1.1	This study
Republic of Tyva	2019	1163	0.36%	1.04:1	[22]
Republic of Tyva	2008	910	0.29%	1:1.4	[20]
Republic of Sakha (Yakutia, Arctic zone)	2022	367	9.2% (of the Momsky District)	1:1.8	[23]
Republic of Sakha (Yakutia)	2018	1045	0.11%	1:1.0	[24]
Republic of Sakha (Yakutia)	2008	982	0.09%	1:1.0	[24]
Khabarovsk Region	2019–2020	4759	0.36%	1:1.1	This study

^1^ Proportion of the study group relative to the officially reported population of the region in the year of sample collection. If the collection spanned two years, data for the first year are shown. ^2^ For age distribution, see Appendix A. ^3^ Abbreviated in the text as (groups of) “St. Petersburg” and “Moscow,” respectively.

**Table 2 viruses-17-01529-t002:** Prevalence of antibodies to the HCV and detection of the HCV RNA in the surveyed regional groups in 2008 and 2018–2022 (M: Males; F: Females). A detailed breakdown by sex and age, including the number of anti-HCV–positive participants in each category, is provided in Appendix A.

Region and Year of Survey	Sex	N	Anti-HCV Positive, % (95% CI)	HCV RNA Positive, % (95% CI)
Kaliningrad Region, 2019	M	518	3.1 ± 1.5	
	F	533	2.8 ± 1.4	
	Combined	1051	2.9 ± 1.0	1.8 ± 0.8
Saint Petersburg and Leningrad Region, 2018–2019	M	1640	2.4 ± 0.7 *	
	F	3680	0.9 ± 0.3 *	
	Combined	5320	1.4 ± 0.3	0.4 ± 0.2
Belgorod Region, 2018	M	858	1.6 ± 0.8 *	
	F	1169	0.7 ± 0.5 *	
	Combined	2027	1.1 ± 0.5	0.8 ± 0.4
Moscow (City and Region), 2018–2019	M	2425	2.2 ± 0.6 *	
	F	5058	0.9 ± 0.3 *	
	Combined	7483	1.3 ± 0.3	0.3 ± 0.1
Moscow (City and Region), 2008	M	486	2.5 ± 1.4	
	F	561	1.1 ± 0.9	
	Combined	1047	1.7 ± 0.8	0.9 ± 0.6
Republic of Dagestan, 2019–2020	M	2635	2.2 ± 0.6 *	
	F	2214	1.3 ± 0.5 *	
	Combined	4843	1.8 ± 0.4	0.6 ± 0.2
Rostov Region, 2008	M	524	2.1 ± 1.2	
	F	430	2.1 ± 1.4	
	Combined	954	2.1 ± 0.9	1.3 ± 0.7
Republic of Tatarstan, 2020	M	294	1.4 ± 1.3	
	F	620	2.4 ± 1.2	
	Combined	914	2.1 ± 0.9	0.9 ± 0.6
Sverdlovsk Region, 2008	M	340	2.6 ± 1.7	
	F	531	3.2 ± 1.5	
	Combined	871	3.0 ± 1.1	0.6 ± 0.5
Novosibirsk Region, 2019–2020	M	4319	2.1 ± 0.4	
	F	4000	1.8 ± 0.4	
	Combined	8319	1.9 ± 0.3	0.5 ± 0.2
Republic of Tyva, 2019	M	594	1.7 ± 1.0	
	F	569	2.3 ± 1.2	
	Combined	1163	2.0 ± 0.8	1.1 ± 0.6
Republic of Tyva, 2008	M	381	3.1 ± 1.8	
	F	529	3.4 ± 1.5	
	Combined	910	3.3 ± 1.2	1.2 ± 0.7
Republic of Sakha (Yakutia), Arctic zone, 2022	M	130	5.4 ± 3.9	
	F	237	5.1 ± 2.8	
	Combined	367	5.2 ± 2.3	1.9 ± 1.4
Republic of Sakha (Yakutia), Southern districts, 2018	M	452	1.8 ± 1.2	
	F	593	2.2 ± 1.2	
	Combined	1045	2.0 ± 0.9	0.6 ± 0.5
Republic of Sakha (Yakutia), Southern districts, 2008	M	402	3.0 ± 1.7	
	F	580	3.4 ± 1.5	
	Combined	982	3.3 ± 1.1	1.2 ± 0.7
Khabarovsk Region, 2019–2020	M	2543	4.2 ± 0.8 *	
	F	2216	2.4 ± 0.6 *	
	Combined	4759	3.4 ± 0.5	1.1 ± 0.3
All regions combined, 2018–2022	M	16,408	2.5 ± 0.2 *	
	F	20,889	1.5 ± 0.2 *	
	Combined	37,291	1.9 ± 0.1	0.6 ± 0.1
All regions combined, 2008	M	2133	2.6 ± 0.7	
	F	2631	2.7 ± 0.6	
	Combined	4764	2.6 ± 0.5	1.0 ± 0.3

* Asterisks (*) indicate statistically significant differences in anti-HCV prevalence between male and female subgroups within the same region (*p*-value < 0.05).

**Table 3 viruses-17-01529-t003:** Pairwise comparison of anti-HCV prevalence between regional groups surveyed in 2018–2022. Each cell at the intersection of a row and a column presents the *p*-value for the difference in anti-HCV prevalence between the corresponding regional groups, calculated using pairwise chi-square tests. Color coding indicates the level of statistical significance: light green, *p* < 0.05; green, *p* < 0.01; “nd”, no statistically significant difference detected. Diagonal cells (highlighted in gray) display the anti-HCV prevalence for that region together with the 95% confidence interval (95% CI).

Regional Groups, 2018–2022	N	n, Anti-HCV (+)	Kaliningrad Region, 2019	Saint Petersburg and Leningrad Region, 2018–2019	Belgorod Region, 2018	Moscow (City and Region), 2018–2019	Republic of Dagestan, 2019–2020	Republic of Tatarstan, 2020	Novosibirsk Region, 2019–2020	Republic of Tyva, 2019	Republic of Sakha (Yakutia), Arctic Zone, 2022	Republic of Sakha (Yakutia), 2018	Khabarovsk Region, 2019–2020
Kaliningrad Region, 2019	1051	31	2.9 ± 1.0%	*p* < 0.01	*p* < 0.01	*p* < 0.01	*p* < 0.05	nd	*p* < 0.05	nd	*p* < 0.05	nd	nd
Saint Petersburg, 2018–2019	5320	72	*p* < 0.01	1.4 ± 0.3%	nd	nd	nd	nd	*p* < 0.05	nd	*p* < 0.01	nd	*p* < 0.01
Belgorod Region, 2018	2027	22	*p* < 0.01	nd	1.1 ± 0.5%	nd	*p* < 0.05	*p* < 0.05	*p* < 0.05	*p* < 0.05	*p* < 0.01	*p* < 0.05	*p* < 0.01
Moscow, 2018–2019 ‖	7483	99	*p* < 0.01	nd	nd	1.3 ± 0.3%	*p* < 0.05	nd	*p* < 0.01	nd	*p* < 0.01	nd	*p* < 0.01
Dagestan, 2019–2020 ǂ	4843	87	*p* < 0.05	nd	*p* < 0.05	*p* < 0.05	1.8 ± 0.4%	nd	nd	nd	*p* < 0.01	nd	*p* < 0.01
Tatarstan, 2020	914	19	nd	nd	*p* < 0.05	nd	nd	2.1 ± 0.9%	nd	nd	*p* < 0.01	nd	*p* < 0.05
Novosibirsk region, 2019–2020 ǂ	8319	159	*p* < 0.05	*p* < 0.05	*p* < 0.05	*p* < 0.01	nd	nd	1.9 ± 0.3%	nd	*p* < 0.01	nd	*p* < 0.01
Tyva, 2019 ‖	1163	23	nd	nd	*p* < 0.05	nd	nd	nd	nd	2.0 ± 0.8%	*p* < 0.01	nd	*p* < 0.05
Yakutia, Arctic zone, 2022 ǂ	367	19	*p* < 0.05	*p* < 0.01	*p* < 0.01	*p* < 0.01	*p* < 0.01	*p* < 0.01	*p* < 0.01	*p* < 0.01	5.2 ± 2.3%	*p* < 0.01	nd
Yakutia, 2018 ‖, ǂ	1045	21	nd	nd	*p* < 0.05	nd	nd	nd	nd	nd	*p* < 0.01	2.0 ± 0.9%	*p* < 0.05
Khabarovsk region, 2019–2020	4759	161	nd	*p* < 0.01	*p* < 0.01	*p* < 0.01	*p* < 0.01	*p* < 0.05	*p* < 0.01	*p* < 0.05	nd	*p* < 0.01	3.4 ± 0.5%
All surveyed regions combined, 2018–2022 ≠	37,291	713											1.9 ± 0.1%

‖, ≠, ǂ Symbols next to the regional group names in the leftmost column denote the following: ‖—no statistically significant difference from the same region in 2008 (see Table 2); ≠—significantly different from the 2008 group (*p* < 0.01); ǂ—significantly different from data for the same region reported by other authors (*p* < 0.01; see details in Section 3.2).

## Data Availability

The data presented in this study are available in the Appendix A. Any additional data are available on request from the corresponding author: victormanuilov@yandex.ru.

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
