# Peer review of "Regional, Age, and Sex Patterns of Hepatitis C Virus Infection in Russia: Insights from a 42,000-Participant Serosurvey"

_viruses, 2025, doi:10.3390/v17121529_

Round 1

Reviewer 1 Report (Previous Reviewer 1)

Comments and Suggestions for Authors

I would like to thank the authors for their careful consideration of the review comments, and the authors have thoroughly addressed the comments from the original submission.  I have no further comments on the manuscript in its current form.

Reviewer 2 Report (Previous Reviewer 2)

Comments and Suggestions for Authors

the paper can be accepted in the present form

Reviewer 3 Report (Previous Reviewer 3)

Comments and Suggestions for Authors

after addressing the queries , the manuscript carries more information

This manuscript is a resubmission of an earlier submission. The following is a list of the peer review reports and author responses from that submission.

Round 1

Reviewer 1 Report

Comments and Suggestions for Authors

Summary

HCV infection descriptive epidemiology is important to understanding the context of HIV prevalence in different populations globally, and it is especially important given the harms associated with untreated HCV infection and the potential cure that is available to those who are positive and untreated.  The authors undertake an analysis of HCV infection in Russia, where the authors note that there are millions of individuals infected with HCV for which there is potential to treat many of these individuals, thereby decreasing the overall rates of morbidity and mortality.  The inclusion of geographical differences is of particular relevance, given the geographical diversity of the different republics of Russia.

Major edits:

In the introduction, the authors note that “population-wide screening seems currently impractical” but do not provide references or support for this statement. Is it due to the large population in Russia? A lack of providers available to conduct testing? The geographic nature of certain areas of Russia for which testing may not be available? Please provide additional context for this statement in the introduction (line  64). I do agree that subpopulation surveillance is the most important (defined by demographics the authors mention).

Further epidemiological description of infection among Russia population would be helpful; are most of these infections that are occurring a result of injection drug use practices or are there other factors at play? (Lines 74-86). For example, in St. Petersburg which has the highest incidence according to the Figure 1 data, there have been studies highlighting injection drug use in this location and evidence of disproportionate rates of HIV and HCV infection (see 10.1097/01.aids.0000218555.36661.9c, for example).

In the methods the authors note that there were 30,724 participants included out of the 42,055 “healthy volunteers”.  I know that in Table 1 that the reasons for these different numbers is the current study vs “first published” participants are added together to make up the total of 42,055.  However, this is not clearly stated in the methods, and should be more specifically mentioned in the first paragraph under section 2.1.  The authors should also make clear how the 8,937 number relates to the other two totals in the paragraph (essentially, the text should read like a flow diagram so that the reader can better understand how the authors came up with the numbers that are presented in the text). 

Where the authors have divided the age groups starting on line 128, it would be helpful to have further rationale as to why some of the age groups are not consistent with respect to the number of years (i.e. some are 5 year age groups while others are 10 year age groups).

In the results, have the authors considered including a figure showing the regions of Russia and HCV prevalence as a map display to help readers contextualize the locations relative to one another? Given the emphasis on showing regional differences in HCV infection, this would be a helpful visual for the reader (particularly those that may not be familiar with the overall regional layout of the different Republics in Russia).

The results divide the regions into three categories of HCV prevalence (low, moderate and high) with ranges of 1.1-1.4, 1.8-2.1, and 3.4-5.2 respectively.  What category would regions fall that have HCV prevalence of 2.2-3.3 (for example, Kaliningrad Region has a prevalence of 2.9; would they be classified as moderate or high?)

On line 229 the authors introduce data for a particular region and give data from a study published for the period 1995-1999. However, in the methods, they make no mention of assessing time points earlier than 2008, which is slightly problematic given that this information shows up “out of the blue” without describing the context of evaluating earlier studies prior to 2008.  While it is perfectly reasonable to do so, the authors should provide context in the methods that they had planned to evaluate other date spans outside of the 2008 or 2018-2022 time periods specified in order to set up the reader for the results presented specific to the Novosibirsk region.  Further results in other regions, of note, also fall outside of the date spans described in the methods (see paragraph starting with line 241).

Starting on line 396 where age disparities are mentioned, the authors could strengthen the sentence by highlighting that it is typically seen that men have a higher prevalence of injection drug use vs women, which is likely one of the dominant explanations for why there is a difference in HCV prevalence among men vs women.

I am a little confused at the age and sex specific calculation that is presented in the paragraph starting with line 463. The authors provide an estimate from the 2019 population of Russia earlier in the results (approximately 147 million) but then provide this new estimate based on a population of only 42 million in 2019. If this is not an error and this is the correct population size estimation for only 1/3 of the population of Russia, then this needs to be explained in the results, and additionally there should be a cautionary note that because this estimate covers a much smaller percentage of Russia, it may not align with what is estimated from the larger population size from the earlier estimate paragraph in the results.

The discussion needs to be reframed or rearranged, such that the authors provide context to the results and interpretations in the data that they present, notably the estimation exercise at the end of the results, before they go into the limitations of their study.  While the limitations are certainly important, they should not be the first topic that is covered in the discussion section of the manuscript.

In Table 2 many of the regions have a higher prevalence of HCV among men vs women, which is expected and the authors do hypothesize some reasons for this trend.  However, there are some regions where females have a higher HCV prevalence; is there any hypothesis to support why this may be the case (one example from Table 2: Republic of Sakha (Yakutia), Southern districts, 2008)

One of the major limitations that I see in the data is the lack of any data presented by the authors for a large time period between the 2008 data and the later data from 2018-2022.  The authors do not make an effort to cite this as a limitation, and the conclusion suggests that data was collected throughout the 14 year period, although this was not the case.  This could have implications for trends that are presented; although there are decreases seen for many regions from 2008 to the 2018-2022 data, it is not known if the trend was a continuous decrease over the 10 year period, or if there were increases or decreases seen during that period that, unfortunately, cannot be assessed was there is no data presented for the 10 year period.

Minor edits:

The authors should check all tables and figure designations in the text, and also check the numbering of tables and figures as they are presented in the paper, as it appears there are some issues with respect to numbering:

Table 1: Please review the Table 1 title; currently it reads: “This is a table. Tables should be placed in the main text near to the first time they are cited.”

Table 2: Consider shortening the overall title name and including more of the detailed information in the Title in footnotes.

Table 2 and what appears to be Table 3 are both labeled as “Table 2” in the manuscript.

There also appears to be several figures designated as “Figure 1” in the manuscript. See page 12 of the manuscript where Figure 1 is designated a second time (also check the text to make sure figure numbers align with the data being presented in the results).

Suggest removing the term “apparently” from line 133 which introduces vague language in the sentence where the authors describe the health of the participants included.  Also, it should be noted that later in the discussion they refer to participants as “conditionally healthy” which is different wording than what appears in other sections where the population is described.

Reviewer 2 Report

Comments and Suggestions for Authors

The paper although well written does not provide relevant information in this relevant field. Molecular/sequencing studies are missing and thus the importance of the results presented is low. We do not reccomend this paper for publication

Reviewer 3 Report

Comments and Suggestions for Authors

Authors have done massive efforts in getting prevalence data for such a large population in Russia. The results have been well described in the manuscript 

The study duration is of 4 years from 2018-2022, unfortunately during 2020-22 COVID-19 struck the world . Authors should clarify regarding the sampling bias during the pandemic period and was sample collected during the period ?

The large data gives opportunity for better understanding the prevalence, from the data it will useful if there was any trend which can help in targeting the population or local practice in higher endemic region which can cause increased prevalence.